# Systematic review and meta-analysis of birth outcomes in women with polycystic ovary syndrome

Mahnaz Bahri Khomami [1] ✉, Somayeh Hashemi[2], Soulmaz Shorakae[1], Cheryce L. Harrison[1,3], Terhi T. Piltonen [4], Daniela Romualdi [5], Chau Thien Tay[1], Aya Mousa[1], Eszter Vanky[6,7] & Helena J. Teede [1,3]

It is unclear whether polycystic ovary syndrome (PCOS) is an independent risk factor for adverse birth outcomes in the offspring of affected women. Here, we investigate the association of PCOS with birth outcomes in the offspring of women with PCOS overall and by potential confounders. This systematic review and meta-analysis included 73 studies and 92,881 offspring of women with and without PCOS from inception until 13th July 2022. We report that mothers with PCOS are younger and have higher body mass index (BMI) around conception and have greater gestational weight gain. The odds of preterm birth, fetal growth restriction and low birth weight are higher and mean birthweight is lower in PCOS of which a lower mean birthweight and a higher small for gestational age are probably independent of BMI. This work informed the recommendations from the 2023 international evidence-based guideline for the assessment and management of polycystic ovary syndrome, emphasizing that PCOS status should be captured at pregnancy to identify risk and improve birth outcomes in the offspring.

Polycystic ovary syndrome (PCOS) impacts ~13% of women during their reproductive years[1] and is diagnosed by oligo/anovulation, clinical and/or biochemical hyperandrogenism, and/or polycystic ovary morphology on ultrasound or elevated anti-Mullerian hormone levels[2]. Women with PCOS are more likely to have anovulatory infertility and undergo fertility treatments, including assisted reproductive technology (ART), to conceive. They are also at a higher risk of obesity[3], insulin resistance[4], diabetes, hypertension, depression, anxiety and poor quality of life[5].

Similar risk factors are seen in pregnant women with PCOS, including higher BMI and gestational weight gain and an increased likelihood of developing gestational diabetes mellitus (GDM) and hypertensive disorders[6–9]. These factors contribute to worsened birth outcomes in offspring[10,11]. Meta-analyses have demonstrated higher preterm birth[7–9,12,13], admission to neonatal intensive care units (NICUs)[7,13] and perinatal mortality[7,9] and lower mean birthweight[7,12,13] in offspring of women with PCOS, compared with offspring of women without PCOS, while fetal growth restriction[9,12], respiratory distress syndrome[9] and neonatal malformations[7,9] appear similar. However, there are inconsistent reports for weight-related indices of offspring of women with and without PCOS (e.g. small for gestational age is reported to be similar[7,9,12] or higher[8] in offspring of women with PCOS). Other key gaps include disparate data regarding the impact of maternal characteristics such as age, BMI, and ART on relevant

[1]Faculty of Medicine, Nursing and Health Sciences, Monash Centre for Health Research and Implementation, Monash University, Melbourne, Australia. [2]Brock University, St. Catharines, ON, Canada. [3]Endocrinology and Diabetes Units, Monash Health, Melbourne, VIC, Australia. [4]Department of Obstetrics and Gynecology, Research Unit of Clinical Medicine, Medical Research Center, Oulu University Hospital, University of Oulu, Oulu, Finland. [5]Department of Woman and Child Health and Public Health, Woman Health Area, Fondazione Policlinico Universitario A. Gemelli, Rome, Italy. [6]Department of Clinical and Molecular Medicine, Faculty of Medicine and Health Sciences, Norwegian University of Science and Technology, Trondheim, Norway. [7]Department of Obstetrics and Gynecology, St. Olav's Hospital, Trondheim University Hospital, Trondheim, Norway. ✉e-mail: mahnaz.bahrikhomami@monash.edu

outcomes, substantial clinical heterogeneity in pooled analyses for multiple outcomes, and insufficient studies precluding subgroup analyses or multivariate or univariate meta-regression[12].

A diagnosis of PCOS can take over 2 years[14], with delays being associated with lower socioeconomic status[15], more severe obesity and higher infertility, hypertension, and dysglycemia, compared with non-PCOS[16,17], all of which adversely impact offspring outcomes. Despite its important implications for pregnancy and offspring health and potential opportunities for improved identification, monitoring and prevention, PCOS is poorly captured in pregnancy and is not widely recognized as a risk factor for adverse birth outcomes[18].

To inform pregnancy recommendations in the upcoming *International Evidence-based Guideline for the Assessment and Management of Polycystic Ovary Syndrome 2023*, we updated our prior systematic review, meta-analysis, and meta-regression to determine the association of PCOS with birth outcomes in offspring of women with and without PCOS. Using a larger number of studies and a broader dataset of participants, we aimed to address the existing evidence gaps regarding the small number of studies per outcome, sensitivity, and risk factor.

## Results

The literature search identified 4595 articles published since 2017, of which 28 studies were included in the meta-analysis after full-text review. Agreements between the reviewers in the title and abstract screening and full-text screening were considered good and excellent, with kappa values of 0.72 and 0.94, respectively[19]. There were 53 studies in the 2017 meta-analysis, of which 45 were eligible in the current review (after excluding studies with PCOS diagnosis by self-report or ICD). Combining the new (n = 28) and previous (n = 45) articles, a total of 73 articles were included in the present systematic review (Fig. 1). Further details are delineated in the Technical Report for the *2023 International Evidence-based Guideline for the Assessment and Management of Polycystic Ovary Syndrome*[20].

The outcomes were reported in 15,070 offsprings of women with PCOS and 77,811 offsprings of women without PCOS (Supplementary Table 1). Thirty-one studies were conducted in Asia, 24 in Europe, 15 in America, one in Australia and New Zealand, and two in Africa. One study recruited women with multiple pregnancies[21]. Five studies reported outcomes in women who took metformin after conception[22-26] and one in women who had conceived after bariatric surgery[27]. Fourteen studies reported outcomes in post-ART pregnancies[28-41] and four reported outcomes in pregnancies with GDM in women with and without PCOS[42-45]. Thirteen studies had a high-quality design[28,38,43,46-55]. Twenty studies matched women with and without PCOS for age[24,25,29,38,43,46-50,52-61] and 14 for age and BMI[38,43,47,49,50,52-54,57-62]. No publication bias was found for any of the outcomes. The certainty of evidence for the outcomes was very low to moderate, mainly due to a high risk of bias, serious inconsistency, and serious indirectness.

Overall, 50 studies reported age[21,23-27,29,33,34,36,38,40-47,50,51,53-56,58-61,63-83], and 45 studies reported BMI[21,23,24,26,27,29,34,36,38,40,41,43-47,50-52,54-61,63-72,75-81,84], either at preconception or at early pregnancy. Women with PCOS were younger (MD: −0.47 years; 95% CI: −0.72, −0.21) and had a higher BMI (1.82 kg/m$^2$; 1.42, 2.22), compared with women without PCOS. Sensitivity analysis showed that after that exclusion of studies in which women were taking metformin after conception[23,24,26], or conceived after bariatric surgery[27], lower age (−0.52 years; −0.78, −0.25) and higher BMI (1.71 kg/m$^2$; 1.30, 2.12) in PCOS remained significant. Fifteen studies reported gestational weight gain[33,43,45,46,50,56,58,59,63,64,67-70,84]. Women with PCOS had higher gestational weight gain compared with women without PCOS (1.06 kg; 0.07, 2.05). Table 1 shows the effect sizes for outcomes of interest on pooled and sensitivity analyses. Forest and cumulative plots, funnel plots (Supplementary Figs. 1a-7c),

and Egger's test results are provided in the Supplementary information.

### Preterm birth

Fifty-six studies reported preterm birth[21,23-31,33-46,48,49,51-56,58-60,63-68,71,74,76-79,82,85-91]; three were excluded from meta-analysis because of overlapping participants[22,53,62], leaving 13,213 offsprings of women with PCOS and 68,830 offsprings of women without, PCOS. Preterm birth was defined as birth prior to 37 weeks of pregnancy in 18 studies[21,27,29,31,36,37,42,45,48,55,59,60,65,68,74,82,89,90], birth prior to 37 weeks of pregnancy after exclusion of indicated deliveries[79], 22–37 weeks in two[33,46], 24–37 weeks in one[38] 28–37 weeks in two[41,76], 32–37 weeks in one[34], and self-reported in one study[28]. The remaining 28 studies did not provide a definition. The odds of preterm birth were comparatively higher in the offspring of women with PCOS (OR: 1.53; 95% CI: 1.33, 1.75). Sensitivity analysis showed that, after the exclusion of studies in which women were taking metformin after conception[23-26] or conceived after bariatric surgery[27], preterm birth remained higher in PCOS (1.57; 1.36, 1.81). Cumulative meta-analysis suggested that the odds of preterm birth in offspring born to women with and without PCOS did not substantially change over time. The higher odds of preterm birth were retained in post-ART pregnancies[28-41], prospective[35,43,45,48,49,51,52,56,60,68,77,86,88,89,91,92] and high-quality studies[28,38,43,46,49,51,52,54,55], but not in pregnancies with GDM[42-45]. In 12 studies, women with and without PCOS were matched for age[29,38,43,46,49,52,54-56,58-60] and in nine for age and BMI[38,43,49,52,54,55,58-60]; PCOS was associated with increased odds of preterm birth in age-matched studies, but not in age- and BMI-matched studies.

### Birthweight

Forty-five studies reported birthweight in 8997 offsprings of women with PCOS and 58,106 offsprings of women without PCOS[21,22,24-27,29,31,33,34,36,37,42-44,46,47,50,53-61,65-74,77,79,83,84,86,93,94]. The offspring of women with PCOS had a comparatively lower mean birthweight (MD: −57.87 g; 95% CI: −97.57, −18.17). Sensitivity analysis after exclusion of studies in women taking metformin after conception[22,24-26] or who conceived after bariatric surgery[27], showed that mean birthweight remained lower in PCOS (−56.02 g; −97.87, −14.16). Cumulative meta-analysis suggested a diminishing magnitude in the MD of birthweight in offspring born to women with and without PCOS did not substantially change over time. The lower mean birthweight was retained in prospective[43,47,53,58,60,61,68-70,72,73,77,86,92,93] and high-quality studies[43,46,47,50,53-55], but not in post-ART pregnancies[29,31,33,34,36,37,95] and in pregnancies with GDM[42-44]. In 14 studies, women with and without PCOS were matched for age[29,43,46,47,50,53-61], and in 11, for age and BMI[43,47,50,53-55,57-61]; PCOS was associated with lower mean birthweight in both age-matched and age- and BMI-matched studies.

### Fetal growth restriction

Thirteen studies reported fetal growth restriction in 2123 offsprings of women with PCOS and 8301 offsprings of women without PCOS[29,48,51,52,54,60,62,67,77,80,82,91,96]. Two studies were excluded from the meta-analysis because of overlapping participants[62,96]. Fetal growth restriction was defined as restriction in fetal growth recognized by a minimum of two ultrasound assessments in one study[48], estimated fetal weight below the 10th percentile for gestational age in one study[82], estimated fetal weight or fetal abdominal circumference below the 10th percentile for gestational age in one study[60], estimated fetal growth indices below the 10th percentile for gestational age in the Chinese population in one study[51], and estimated fetal weight below 1.5 SD in the Japanese population in one study[54]. The remaining six studies did not provide a definition. The odds of fetal growth restriction were higher in the offspring of women with PCOS compared with the offspring of women without PCOS (OR: 1.84; 95% CI: 1.09, 3.10). There were no studies in which women were taking metformin after conception or

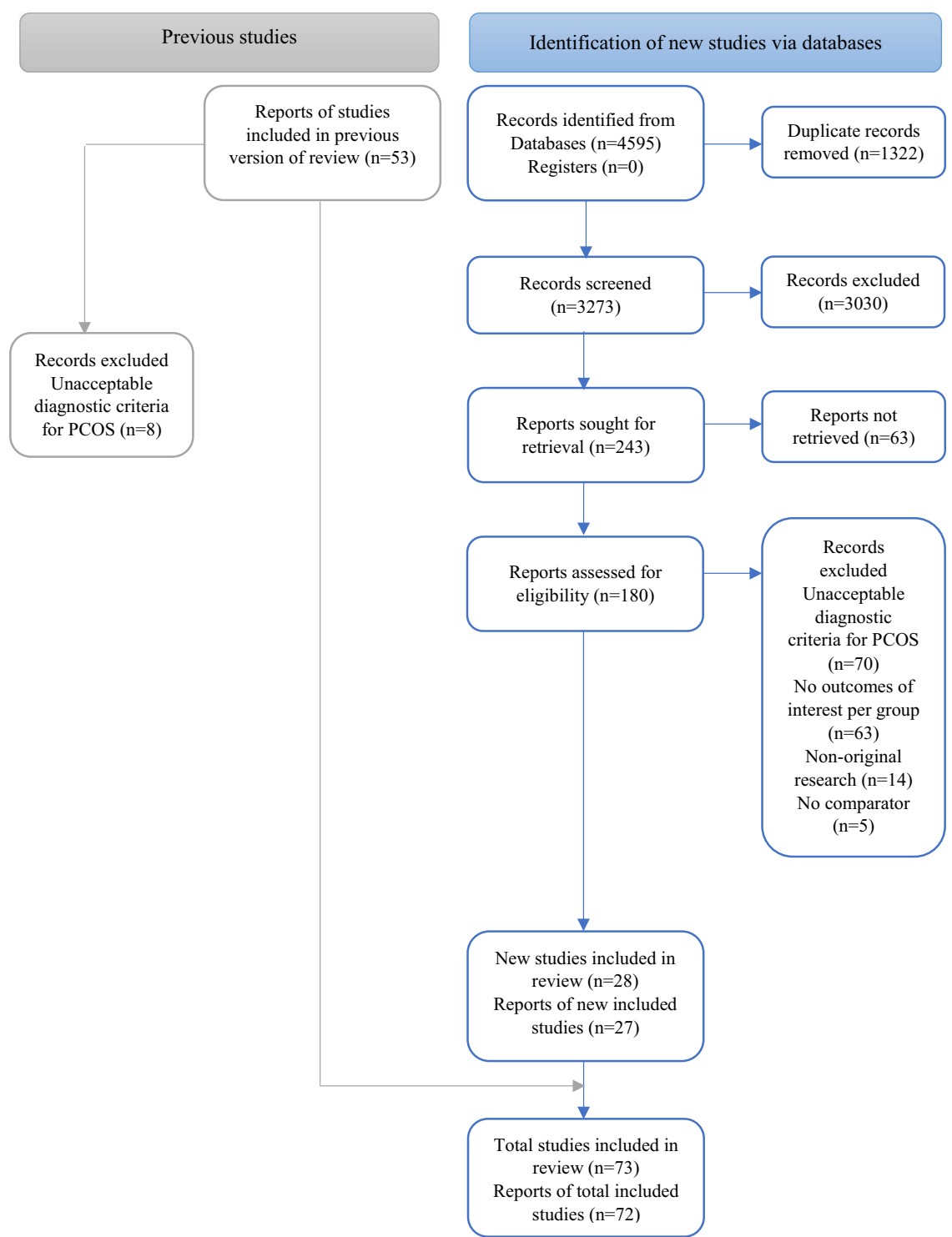

**Fig. 1 | PRISMA flow diagram of study selection.** This figure illustrates the PRISMA (Preferred Reporting Items for Systematic Reviews and Meta-Analyses) flow diagram detailing the study selection process. The diagram includes the number of records identified, screened, assessed for eligibility, and included in the meta-analysis.

conceived after bariatric surgery. Convergence could not be achieved during tau² estimation in random-effects cumulative meta-analysis. The higher odds of fetal growth restriction were retained in prospective studies[51,53,60,62,77,91] but were not retained in post-ART pregnancies[29] and high-quality studies[51,52,54]. No studies reported fetal growth restriction in pregnancies with GDM. In four studies, women with and without PCOS were matched for age-[29,52,54,60] and in three for age and BMI[52,54,60]; PCOS was associated with fetal growth restriction neither in age-matched studies nor in age- and BMI-matched studies.

**Low birth weight**

Fifteen studies reported low birth weight in 7079 offsprings of women with PCOS and 42,848 offsprings of women without PCOS[21,27,30,33,34,36,37,39,40,55,64,68,74,76,78]. Low birth weight was defined as birthweight below 2500 g at the 37th week of pregnancy in three studies[30,34,68], and birthweight below 2500 g in nine studies[21,27,33,36,37,40,55,74,76]. The remaining three studies did not provide a definition. The odds of low birth weight were higher in the offspring of women with PCOS compared with the offspring of women without

## Table 1 | Association of maternal polycystic ovary syndrome with offspring outcomes

| Outcome | No. of studies | Effect size [95% CI] | $I^2$ (%) |
|---|---|---|---|
| **Preterm birth** | 54 | OR 1.53 [1.33, 1.75] | 67.6 |
| No metformin/bariatric surgery | 49 | OR 1.57 [1.36, 1.81] | 68.7 |
| Post ART pregnancies | 14 | OR 1.46 [1.14, 1.87] | 80.7 |
| Prospective design | 16 | OR 1.67 [1.28, 2.20] | 40.9 |
| Pregnancies with GDM | 4 | OR 0.90 [0.58, 1.41] | 0.0 |
| High quality | 9 | OR 1.89 [1.12, 3.20] | 56.2 |
| Age[a] matched | 12 | OR 1.57 [1.08, 2.29] | 25.1 |
| BMI[b] matched | 9 | OR 1.58 [0.92, 2.71] | 41.5 |
| **Birthweight (g)** | 45 | MD −57.87 [−97.57, −18.17] | 87.1 |
| No metformin/bariatric surgery | 40 | MD −56.02 [−97.87, −14.16] | 88.8 |
| Post ART pregnancies | 7 | MD 13.44 [−35.21, 62.08] | 65.6 |
| Prospective design | 15 | MD −93.44 [−178.35, −8.53] | 93.6 |
| Pregnancies with GDM | 3 | MD −35.31[−133.45, 62.84] | 0.0 |
| High quality | 7 | MD −139.09 [−219.07, −59.11] | 44.0 |
| Age matched | 14 | MD −109.46 [−170.29, −48.62] | 43.5 |
| BMI matched | 11 | MD −101.20 [−167.57, −34.84] | 48.9 |
| **Fetal growth restriction** | 11 | OR 1.84 [1.09, 3.10] | 37.2 |
| No metformin/bariatric surgery | 11 | OR 1.84 [1.09, 3.10] | 37.2 |
| Post ART pregnancies | 1 | OR 1.74 [0.47, 6.23] | . |
| Prospective design | 6 | OR 3.37 [1.57, 7.25] | 18.7 |
| Pregnancies with GDM | 0 | – | – |
| High quality | 3 | OR 2.40 [0.60, 9.61] | 68.2 |
| Age matched | 4 | OR 1.38 [0.58, 3.28] | 0.0 |
| BMI matched | 3 | OR 1.16 [0.37, 3.67] | 0.0 |
| **Low birth weight** | 15 | OR 1.28 [1.04, 1.59] | 58.6 |
| No metformin/bariatric surgery | 14 | OR 1.27 [1.03, 1.57] | 59.6 |
| Post ART pregnancies | 7 | OR 1.37 [1.03, 1.81] | 66.0 |
| Prospective design | 1 | OR 1.57 [0.92, 2.65] | . |
| Pregnancies with GDM | 0 | – | – |
| High quality | 1 | OR 1.36 [0.58, 3.18] | . |
| Age matched | 0 | – | – |
| BMI matched | 0 | – | – |
| **Small for gestational age** | 25 | OR 1.11 [0.87, 1.41] | 52.7 |
| No metformin/bariatric surgery | 24 | OR 1.13 [0.88, 1.45] | 53.9 |
| Post ART pregnancies | 3 | OR 0.70 [0.51, 0.97] | 0.0 |
| Prospective design | 12 | OR 1.78 [1.18, 2.67] | 56.0 |
| Pregnancies with GDM | 2 | OR 1.23 [0.52, 2.93] | 0.0 |
| High quality | 4 | OR 2.21 [0.87, 5.65] | 69.8 |
| Age matched | 5 | OR 2.91 [1.37, 6.19] | 53.5 |
| BMI matched | 5 | OR 2.91 [1.37, 6.19] | 53.5 |
| **Macrosomia** | 23 | OR 1.14 [0.95, 1.35] | 61.0 |
| No metformin/bariatric surgery | 22 | OR 1.16 [0.97, 1.39] | 60.4 |
| Post ART pregnancies | 6 | OR 1.16 [0.77, 1.75] | 88.2 |
| Prospective design | 5 | OR 1.17 [0.83, 1.64] | 0.0 |
| Pregnancies with GDM | 4 | OR 1.37 [0.89, 2.10] | 0.0 |
| High quality | 3 | OR 0.95 [0.65, 1.38] | 0.0 |
| Age matched | 2 | OR 1.24 [0.34, 4.60] | 0.0 |
| BMI matched | 2 | OR 1.24 [0.34, 4.60] | 0.0 |
| **Large for gestational age** | 24 | OR 1.14 [0.98, 1.33] | 18.1 |
| No metformin/bariatric surgery | 22 | OR 1.16 [0.99, 1.35] | 17.2 |
| Post ART pregnancies | 2 | OR 1.54 [0.56, 4.19] | 84.3 |

## Table 1 (continued) | Association of maternal polycystic ovary syndrome with offspring outcomes

| Outcome | No. of studies | Effect size [95% CI] | $I^2$ (%) |
|---|---|---|---|
| Prospective design | 13 | OR 1.24 [0.97, 1.58] | 5.9 |
| Pregnancies with GDM | 2 | OR 1.24 [0.64, 2.40] | 0.0 |
| High quality | 4 | OR 1.23 [0.73, 2.07] | 36.5 |
| Age matched | 5 | OR 1.52 [0.95, 2.43] | 0.0 |
| BMI matched | 5 | OR 1.52 [0.95, 2.43] | 0.0 |

*ART* assisted reproductive technology, *BMI* body mass index, *GDM* gestational diabetes mellitus, *MD* mean difference, *OR* odds ratio.
[a]Age is reported in years.
[b]BMI is reported in kg/m².

PCOS (OR: 1.28; 95% CI: 1.04, 1.59). Sensitivity analysis showed that after the exclusion of one study after bariatric surgery[27], low birth weight remained higher in PCOS (1.27; 1.03, 1.57). Cumulative meta-analysis suggested that the odds of low birth weight in offspring born to women with and without PCOS did not substantially change over time; however, one study published before 2006 contributed to heterogeneity. The higher odds of low birth weight were retained in post-ART pregnancies[30,33,34,36,37,39,40], but not in prospective[68] and high-quality studies[55]. No studies reported low birth weight in pregnancies with GDM. There were no studies where women with and without PCOS were matched for age and/or BMI.

### Small for gestational age
Twenty-seven studies reported small gestational age in 3989 off-springs of women with PCOS and 25,129 offsprings of women without PCOS[21,25,30,33,37,42,48,52,55,58,62,69–71,73–75,80,87–89,96]. Two studies were excluded from the meta-analysis because of overlapping participants[62,96]. Small for gestational age was defined as birthweight below the 5th percentile for gestational age in two studies[58,69], birthweight below the 10th percentile for gestational age in 14 studies[25,29,30,33,42,55,59,70,73–75,80,87,89], birthweight below the 10th percentile for gestational age and sex in the Chinese population in one study[37], birthweight below the 22nd percentile for gestational age and sex in two studies[21,88], and fetal indices below the 5th percentile for gestational age in the white population in one study[48]. The remaining nine studies did not provide a definition. The odds of small for gestational age were similar in offspring of women with and without PCOS (OR: 1.11; 95% CI: 0.87, 1.41), including in sensitivity analysis after excluding one study in which women were taking metformin after conception[25] (1.13; 0.88, 1.45). Cumulative meta-analysis suggested a diminishing magnitude in the odds of small for gestational age in offspring born to women with and without PCOS over time. The odds of small for gestational age were lower in offspring of women with PCOS in post-ART pregnancies[30,33,37] and prospective studies[43,49,52,58,62,69,70,72,73,86,88,89], but not in high-quality studies[43,49,53,55] and in pregnancies with GDM (*P* > 0.05). Two studies reported small for gestational age, the odds of which were similar in the offspring of women with and without PCOS (1.23; 0.52, 2.93). In five studies, women with and without PCOS were matched for both age[43,49,53,58,59] and BMI[43,49,53,58,59]; PCOS was associated with increased odds of small for gestational age in these studies.

### Macrosomia
Twenty-three studies reported macrosomia in 9078 offsprings of women with PCOS and 53,411 offsprings of women without PCOS[22,30,33,34,36,37,40,42–45,51,55,60,64,65,67,68,74,76,78,82,94]. Macrosomia was defined as birthweight above 4000 g in nine studies[22,33,36,37,40,42,45,55,82], birthweight equal to or above 4000 g in three studies[51,74,76], birthweight above 4000 g at the 37th week of pregnancy in three studies[30,34,68], birthweight above 4000 g or above the 95th percentile for gestational age in one study[68], and birthweight above 4500 g in one study[65]. The

remaining six studies did not provide a definition. The odds of macrosomia were similar in the offspring of women with and without PCOS (OR: 1.13; 95% CI: 0.95, 1.35; $I^2$: 61.0%), and this was unchanged in sensitivity analysis after the exclusion of one study in which women were taking metformin after conception[22] (1.16; 0.97, 1.39). Cumulative meta-analysis suggested that the odds of macrosomia in offspring born to women with and without PCOS did not substantially change over time. The odds of macrosomia remained similar in post-ART pregnancies[30,33,34,36,37,40], prospective[43,45,51,60,68], and high-quality studies[43,51,55], and in pregnancies with GDM[42–45]. In two studies, women with and without PCOS were matched for both age and BMI[43,60]; PCOS was not associated with macrosomia in these studies.

### Large for gestational age
Twenty-five studies reported large for gestational age in 1029 offsprings of women with PCOS and 10,052 offsprings of women without PCOS[25,30,37,42,43,48,49,53,55,58,59,62,69–75,77,80,86,88,89,96]. Two studies were excluded from the meta-analysis because of overlapping participants[62,96]. Large for gestational age was defined as birthweight above the 90th percentile for gestational age in fourteen studies[25,30,42,55,58,59,69,70,73–75,80,87,89], birthweight above the 90th percentile for gestational age and sex in the Chinese population in one study[37], and fetal indices above the 95th percentile for gestational age in the white population in one study[48]. The remaining seven studies did not provide a definition. The odds of being large for gestational age were similar in the offspring of women with and without PCOS (OR: 1.14; 95% CI: 0.98, 1.33). Odds of large for gestational age remained similar in sensitivity analysis excluding a study in which women were taking metformin after conception[25] (1.16; 0.99, 1.35), and in post-ART pregnancies, prospective[43,48,49,53,58,69,70,72,73,77,86,88,89] and high-quality studies[43,49,53,55] and in pregnancies with GDM[42,43]. Cumulative meta-analysis suggested a diminishing magnitude in the odds of large for gestational age in offspring born to women with and without PCOS over time. In five studies, women with and without PCOS were matched for both age and BMI[43,49,53,58,59]; PCOS was not associated with large for gestational age in these studies.

### Meta-regression
Significant heterogeneity ($I^2 > 50\%$) was observed for studies reporting preterm birth, birthweight, low birth weight, small for gestational age and macrosomia. Where reported (Table 2), maternal age was lower in PCOS for preterm birth and birthweight; gestational weight gain was higher in PCOS for birthweight; and BMI was higher in PCOS for preterm birth, birthweight, low birth weight, small for gestational age and macrosomia. Variations in age, BMI, and gestational weight gain were not associated with increased odds/lower MD for any of the outcomes on meta-regression.

## Discussion
In this systematic review and meta-analysis of 73 published articles in 92,881 offsprings of women with and without PCOS, women with PCOS were younger in age and had higher BMI around conception and higher gestational weight gain. The offspring of women with PCOS were more likely to be born preterm and have lower mean birthweight, low birth weight, and fetal growth restriction. Preterm birth and low birth weight remained higher in the offspring of women with PCOS in post-ART pregnancies. Higher preterm birth and lower mean birthweight also remained associated with maternal PCOS in both prospective and high-quality studies. In subgroup meta-analyses of studies matched for maternal age or BMI, PCOS was associated with lower mean birthweight and higher small for gestational age but was not associated with fetal growth restriction. Given the higher risks of adverse offspring health outcomes in PCOS, there is a clear need for greater PCOS recognition and identification and for fetal monitoring to facilitate prevention, especially in post-ART pregnancies and pregnancies with higher maternal BMI. This has led to new recommendations in the International PCOS Guideline that PCOS status should be identified during pregnancy to provide appropriate monitoring and support.

Advanced maternal age at pregnancy is associated with higher preterm birth and fetal growth restriction in offspring as well as higher obesity, infertility, pregnancy complications, and chronic diseases in women[97], which further exacerbate birth outcomes in offspring[11]. We found that mothers with PCOS had a higher BMI around conception and had higher gestational weight gain, despite being slightly younger than mothers without PCOS. These have been assessed and consistently reported in just one[12] of the five[7–9,13] previous systematic reviews. While meta-analyses of study-level aggregate data do not allow for evaluation of confounding effects or independent associations, exploring subsets of studies matched for, or limited to, confounding variables, as well as adjusting for confounders, may provide

**Table 2 | Effect of maternal age, BMI and gestational weight gain on the association of maternal polycystic ovary syndrome with offspring outcomes**

| Outcomes | Age (year) | | | BMI (kg/m²) | | | Gestational weight gain (kg) | | |
|---|---|---|---|---|---|---|---|---|---|
| | N | Effect size | Tau² | N | Effect size | Tau² | N | Effect size | Tau² |
| **Preterm birth** | | | | | | | | | |
| MD [95%CI] | 32 | −0.49 [−0.85, −0.13] | – | 28 | 1.71 [1.23, 2.20] | – | 11 | 0.86 [−0.14, 1.86] | – |
| Univariate coefficient (95% CI) | 33 | 0.16 [−0.02, 0.34] | 0.13 | 28 | 0.11 [−0.02, 0.24] | 0.14 | 11 | 0.11 [−0.17, 0.39] | 0.10 |
| **Birthweight (g)** | | | | | | | | | |
| MD [95%CI] | 33 | −0.45 [−0.84, −0.054] | – | 30 | 1.52 [0.87, 2.16] | – | 12 | 1.32 [0.16, 2.47] | – |
| Univariate coefficient (95% CI) | 33 | −0.01 [−0.21, 0.19] | 0.03 | 30 | 0.04 [−0.09, 0.17] | 0.03 | 12 | −0.09 [−0.20, 0.03] | 0.11 |
| **Low birth weight** | | | | | | | | | |
| MD [95%CI] | 11 | −0.40 [−1.01, 0.22] | – | 9 | 1.32 [0.73, 1.91] | – | 3 | −0.55 [−1.90, 0.80] | – |
| Univariate coefficient (95% CI) | 11 | 0.24 [−0.11, 0.59] | 0.14 | Insufficient number | | – | 3 | Insufficient number | – |
| **Small for gestational age** | | | | | | | | | |
| MD [95%CI] | 15 | −0.47 [−1.06, 0.13] | – | 10 | 1.95 [0.14, 3.75] | – | 6 | 0.50 [−0.86, 1.87] | – |
| Univariate coefficient (95% CI) | 15 | 0.05 [−0.23, 0.34] | 0.03 | 10 | −0.02 [−0.25, 0.21] | 0.17 | 6 | Insufficient number | – |
| **Macrosomia** | | | | | | | | | |
| MD [95%CI] | 19 | −0.25 [−0.69, 0.188] | – | 15 | 1.78 [1.17, 2.40] | – | 6 | 0.61 [−0.63, 1.85] | – |
| Univariate coefficient (95% CI) | 19 | −0.05 [−0.25, 0.15] | 0.07 | 15 | −0.03 [−0.22, 0.15] | 0.07 | 6 | Insufficient number | – |

*BMI* body mass index, *MD* mean difference.

an indication of the impact of confounders. Here, in the majority of sensitivity analyses, the odds of adverse birth outcomes increased in the offspring of women with PCOS in age- or BMI-matched studies and decreased in post-ART pregnancies, with variable changes in the direction of effects. The odds of adverse birth outcomes were similar in the offspring of women with and without PCOS in pregnancies with GDM. However, our sensitivity analyses lacked power due to the small number of studies per analysis and/or moderate to substantial heterogeneity[98]. On meta-regression, we did not find a confounding effect from variations in age, BMI, or gestational weight gain across studies in odds ratios. This was in contrast with our findings in the sensitivity analysis for preterm birth, where the increased odds appeared to be related to maternal BMI. An individual patient data meta-analysis may be able to distinguish different characteristics between pregnant women and whether they mediate the association of PCOS with adverse birth outcomes in offspring.

Globally, 10.6% of offspring are born preterm, with a high burden of morbidity in both mothers and offspring, as well as an increased neonatal mortality[99]. Maternal low socioeconomic status, age and BMI extremes, poor diet, overdistension of the uterus, pregnancy complications, chronic disease, and increased inflammatory markers are risk factors for preterm birth[100]. Here, we found that the offspring of women with PCOS have 53–57% higher odds of preterm birth, independent of time and ART. In both prospective and high-quality studies, the association of PCOS with preterm birth was retained. While these findings are consistent with our prior systematic review[12], here, we found for the first time that the higher odds of preterm birth were in offspring of younger women with PCOS[100]. Women with PCOS have chronic inflammation that is independent of but worsened by obesity[101]. In early pregnancy, a strong mobilization of inflammatory and other cytokines has been reported in PCOS, persisting throughout pregnancy, indicating a more activated immune status[83]. Women with PCOS also have higher weight gain[102], and higher GDM and hypertensive disorders of pregnancy independent of, but worsened by, aging and obesity[6]; thus, our finding of preterm birth being reported in offspring of women with PCOS at a younger age and with a higher BMI, bodes potentially higher odds of preterm birth in offspring of an older population of women with PCOS.

Worldwide, ~14.6% of offspring are born at low birth weight with significant short- and long-term sequelae[103]. Maternal age extremes, multiple pregnancies, pregnancy complications and chronic diseases, infections and poor diet are risk factors for a lower birthweight. We found that the offspring of women with PCOS have a 41.52 g lower mean birthweight and 28% higher odds of low birth weight, regardless of sex and gestational age at birth. Overlapping measures are fetal growth restriction and small for gestational age. While fetal growth restriction reflects poor intrauterine growth mostly resulting from placental insufficiency, small for gestational age, which takes fetal gestational age and sex into account, does not indicate the cause of the finding (i.e. genetic/epigenetic variations, insufficient placenta, inadequate nutrition)[104]. We found that, while the offspring of women with PCOS had higher odds of fetal growth restriction, the odds of small for gestational age were similar to the offspring of women without PCOS at birth. These disparate findings could be due to the timing, diagnostic measurements used, and accuracy of ultrasonographic assessments of offspring growth[105].

Macrosomia, with a prevalence of 5–15%, is associated with maternal age, obesity, gestational weight gain, GDM, and induced and operative delivery[106]. We found that the offspring of women with and without PCOS had similar odds of macrosomia and large for gestational age. While our findings confirm the results of our previous systematic review[12], this contrasts with our findings for large for gestational age, which was similar here but higher in PCOS in the prior systematic review[12]. Given that women with PCOS usually have higher mean BMI gestational weight gain and rates of GDM than their non-

PCOS counterparts[6], higher odds of macrosomia and large for gestational age in offspring of women with PCOS were expected. Our current findings on similar odds of macrosomia and large for gestational age in offspring of women with and without PCOS, despite higher BMI, weight gain, and GDM as known risk factors, could be related to opposing effects of the mechanisms driving fetal growth restriction and low birth weight in PCOS, including placental insufficiency[104]. Over time, the odds of small for gestational age diminished in PCOS. This could be due to later studies including younger women with PCOS[107] who are less likely to have chronic conditions[108].

The strengths of this systematic review and meta-analysis include a large number of observational studies and participants across five continents and the exclusion of less reliable PCOS diagnoses. The inclusion of a reliable diagnosis minimizes inaccuracies due to recall bias or misinterpretation of the diagnosis and the inherent variability in diagnostic criteria across medical organizations[109,110]. For the first time, we reported mean age and BMI, corresponding to each outcome of interest. We performed sensitivity and cumulative analyses to examine the potential impact of confounders and performed meta-regression to assess the impact of age, BMI, and gestational weight gain where substantial heterogeneity was observed, a pioneering approach from our previous systematic review. Certainty of evidence was assessed for individual outcomes using the GRADE approach. Limitations should also be noted. A limitation of this review is the inclusion of studies that were published in English, which contributes to potential language bias since English studies are more likely to demonstrate positive findings. Approximately 82% of the included studies had a moderate to high risk of bias, mainly due to high confounding bias (53.4%), followed by high selection bias (38.35%). However, sensitivity analyses of high-quality studies confirmed the overall findings. The majority of studies did not report gestational weight gain, and many had unclear or inconsistent definitions for some of the outcomes. A small number of studies were available for subgroup analyses by age-matched or BMI-matched cohorts, post-ART pregnancies, pregnancies with GDM, or high-quality studies. As this is a meta-analysis of aggregate level data, we were unable to account for confounders in our analyses and could only explore this in studies matched by age, BMI, or limited to a post-ART population or pregnancies with GDM.

In this systematic review and meta-analysis, we found that offspring of women with PCOS are more likely to have fetal growth restriction and be born preterm, with low birth weight. This systematic review and meta-analysis of 92,881 infants of women with and without PCOS directly inform the 2023 International Evidence-based PCOS Guideline recommendation to consider PCOS as a risk factor for adverse birth outcomes in preconception and antenatal guidelines to facilitate screening, proper monitoring, and timely intervention. Other risk factors, such as age, BMI, and ART, should also be considered in women with PCOS. The recommendations outlined in the PCOS guideline hold the potential to influence offspring born to 17 million pregnancies with PCOS annually across the globe.

Additional research, by means of individual patient data meta-analyses with enough statistical power, is warranted to further clarify the impact of PCOS features and associated conditions on birth outcomes in the offspring.

## Methods

### Search strategy and selection criteria

This systematic review and meta-analysis represents an updated version of previously published systematic reviews[6,12] and follows the Meta-Analyses and Systematic Reviews of Observational Studies (MOOSE) guidelines[111]. Given that the majority of the studies on birth outcomes in prospectively identified PCOS come from selected populations with a high-risk profile[112], we prioritized the overall quality of the studies over the distinction between retrospective and

prospective study designs. The protocol for the original review was prospectively registered in PROSPERO (CRD 42017067147). Methods and findings of the previous search, which included publications up to the 4 April 2017, were previously published[6,12]. For this update, the search was limited to English language studies and updated (M.B.K.) from 2017 to 13 July 2022 through Medline, Medline in-process, and other non-indexed citations, EMBASE, and all EBM reviews, including Cochrane Database of Systematic Reviews, Cochrane Clinical Answers, Cochrane Central Register of Controlled Trials, American College of Physicians Journal Club, Cochrane Methodology Register, Health Technology Assessments, The Database of Abstracts of Reviews of Effectiveness and the National Health Service Economic Evaluation Database. Bibliographies of relevant systematic reviews and meta-analyses were searched to identify any additional studies. The search incorporated a broader range of outcomes and was distributed across two systematic reviews and meta-analyses.

Eligible studies included studies reporting observational data on preterm birth, birthweight, fetal growth restriction, low birth weight, small for gestational age, macrosomia, and large for gestational age in offspring of both women with and without PCOS. Our search was limited to studies published in English. A protocol amendment was made to only include studies where the PCOS diagnosis met the Rotterdam criteria[110]. Birth outcomes were accepted based on the definitions provided in the primary studies.

Studies published in languages other than English, case reports, case series, editorials, scoping, and narrative reviews were excluded. Additionally, we excluded studies using self-reported or International Classification of Diseases (ICD) for PCOS diagnosis, and studies not reporting the outcomes of interest in the two groups of PCOS versus non-PCOS.

Two reviewers (M.B.K., S.H., or S.S.) independently screened studies by titles and abstracts and reviewed full texts for eligibility. Discrepancies were discussed and resolved through consensus or arbitration between reviewers. Studies identified through the new search were combined with those published before April 2017. Studies not meeting the new criteria needed for PCOS diagnosis were excluded (i.e. studies captured in the 2017 reviews[6,12] that used self-report or ICD for PCOS diagnosis were excluded from the current review).

### Data analysis
Data extraction and quality appraisal were independently performed by the same reviewers. Using an a priori researcher-developed data extraction form, data were extracted from each study. These included the author, year of publication, study design, study location, participant characteristics, and frequency/mean and standard deviation of outcomes per group. Participant characteristics included maternal age, BMI, mode of conception, existing medical conditions, and medications used during pregnancy.

When participants were overlapping between multiple publications for the same outcome, data from the study with the largest sample size were included in the meta-analysis.

Risk of bias was independently assessed by two reviewers (M.B.K., S.H., or S.S.) using the Newcastle–Ottawa Scale (NOS) for non-randomized studies for selection, comparability, and outcome ascertainment[113]. Studies were considered high-quality if they scored at least three stars in selection, one star in comparability, and two stars in outcome ascertainment. Studies with two stars in selection, a minimum of one star in comparability, and two stars in outcome ascertainment were considered fair-quality. Studies meeting neither of these two thresholds were considered low-quality.

We performed random-effects meta-analyses to generate pooled effect estimates, reported as the mean differences (MDs) or odds ratios (ORs) with 95% confidence intervals (CIs), for the association between PCOS status and pregnancy outcomes. Between-study heterogeneity was assessed using the $I^2$ statistic; $I^2$ above 50% was deemed substantial heterogeneity[114].

Sensitivity analyses were performed excluding studies in which women were taking metformin after conception or conceived after bariatric surgery. Prior research indicates that metformin intake in pregnancy[115] and conception post bariatric surgery[116] may impact birth outcomes in offspring. Cumulative random-effects meta-analyses based on Hedges's adjusted $g$ and its 95% confidence intervals (95% CI)[117] were performed to explore the effect of time (publication year) on the association of PCOS with each outcome of interest. Publication bias was assessed using funnel plots. We assessed the certainty of evidence for each outcome according to the Grading of Recommendations, Assessment, Development and Evaluation (GRADE) system[118] using Gradepro software[119].

Sensitivity analyses of the outcomes by conception with ART (in vitro fertilization, in vitro maturation intracytoplasmic sperm injection, zygote intrafallopian transfer, and gamete intro-fallopian transfer), pregnancies with GDM, and prospective and high-quality studies as well as by age- and BMI-matched designs were also performed after exclusion of studies in which women were taking metformin after conception or conceived after bariatric surgery.

Restricted maximum likelihood (REML)-based random effects meta-regression[120] was performed to explore the effects of variations in maternal age, BMI, and gestational weight gain on each outcome of interest if there were at least 10 studies per coefficient. Effect sizes of age, BMI, and gestational weight gain for individual studies were used in meta-regression analyses. The percentage of between-study variance explained by the model (tau²) was estimated using Knapp–Hartung modification. Normal distributions for mean values were checked using skewness-kurtosis tests. As age and BMI were not both significant ($p < 0.01$) for any of the outcomes on univariate meta-regression analyses, multivariable meta-regression could not be performed. Statistical significance was defined as two-sided $p < 0.05$. All statistical analyses were performed using Stata version 17 (StataCorp, 14 College Station, TX, USA).

### Reporting summary
Further information on research design is available in the Nature Portfolio Reporting Summary linked to this article.

## Data availability
Data that support the findings of this study have been deposited in https://data.mendeley.com/datasets/npy96r94p2/1 (https://doi.org/10.17632/npy96r94p2.1). Source data are provided with this paper.

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

## Acknowledgements

We extend our gratitude to the colleagues who contributed to the initial systematic review published in 2019. While these collaborators were not directly involved in this update, their contributions to the foundational work have been instrumental in shaping the direction and scope of the current study. This work is supported by the Australian National Health and Medical Research Council (NHMRC) funded Centre for Research Excel-lence in Women's Health in Reproductive Life (CRE-WHiRL) [APP#1171592] (H.J.T.). The funder of the study had no role in study design, data extrac-tion, data analysis, data interpretation, writing of the manuscript, or the decision to submit the manuscript for publication.

## Author contributions

M.B.K. contributed to the concept and design, and performed the systematic search and statistical analysis. M.B.K., S.S., and S.H. performed screening, data extraction, quality appraisal, and drafting of the paper. H.J.T. obtained funding and led the overarching guideline process. C.T.T. and A.M. were evidence synthesis leads for the guidelines included in this study. E.V. was the clinical expert lead in the guideline committee. M.B.K., S.S., S.H., C.H.L., T.T.P., D.R., C.T.T., H.J.T., E.V., and A.M. contributed to the concept and design and provided substantial contributions to the drafting of the work, including critical revision for important intellectual content. M.B.K., S.S., S.H., C.H.L., T.T.P., D.R., C.T.T., H.J.T., E.V., and A.M. had full access to the data, were responsible for study integrity, and approved the final version of the manuscript for publication.

## Competing interests

T.P. is supported by Novo Nordisk and Sigrid Jusélius Foundation. E.V. is supported by Novo Nordisk and Merck as a lecturer and advisor for clinical studies. H.J.T. and A.M. are supported by NHMRC fellowships [APP#2009326 and APP#1161871, respectively]. The remaining authors declare no competing interests.

## Inclusion and ethics

All researchers involved in the study are listed as authors or acknowledged, as appropriate.
