## [Peer Review File · Nature Communications]

REVIEWERS' COMMENTS

Reviewer #4 (Remarks to the Author):

I have reviewed the manuscript before. The authors answered my concerns. I have nothing further.

Reviewer #5 (Remarks to the Author):

The searches were conducted in April 2017 and subsequently updated in July 2022. It is highly recommended that the authors extend the review to encompass studies up until 2024 to ensure the most current data informs the analysis.

There is a notable deviation from the original study design outlined in the PROSPERO protocol. For instance, while the protocol anticipates the inclusion of RCTs, these were not considered in the final manuscript. Additionally, the primary outcome of prevalence, as stated in the protocol, is absent from the manuscript, calling into question the protocol's relevance to the study. The authors should revise their PROSPERO protocol to accurately reflect the changes made, detailing the rationale behind these modifications and the strategies employed to mitigate any resultant limitations.

The authors are encouraged to reformat the presentation of results in the supplementary materials. The current approach of directly inserting Stata-generated outputs is suboptimal. Improved legibility, particularly concerning the font size in the supplementary figures, would greatly enhance the readability and interpretability of the data.

The inclusion of studies published between 2003 and 2019 (n=22) appears to contribute to heterogeneity; how might the impact of COVID-19—such as infection during pregnancy or restricted access to healthcare during lockdowns—affect these findings? If this has not been considered, the authors should investigate the potential influences of the pandemic on the study outcomes.

We thank the editor and reviewers for their constructive feedback. We have carefully revised our manuscript, providing point-by-point responses to each comment below.

REVIEWERS' COMMENTS

Reviewer #4 (Remarks to the Author):

I have reviewed the manuscript before. The authors answered my concerns. I have nothing further.

Response: Thanks.

Reviewer #5 (Remarks to the Author):

The searches were conducted in April 2017 and subsequently updated in July 2022. It is highly recommended that the authors extend the review to encompass studies up until 2024 to ensure the most current data informs the analysis.

Response: We appreciate your request for an updated search to ensure the relevance and currency of our systematic review. The initial search was conducted with meticulous attention to detail and rigor, and the results have informed a recent guideline that is pivotal for current clinical practice. Due to the lengthy peer review process, our search may appear outdated. However, the findings of our review are integral to the recent guideline, and updating the search at this point would necessitate a re-evaluation of the entire dataset and potentially delay the translation of these critical guidelines into practice.

Our team has carefully considered the implications of updating the search. Given the significance of the timely publication of these results to support the recent guideline, maintaining the current dataset without an update is in the best interest of advancing public and clinical health imperatives.

Moreover, the current results remain robust and relevant, providing a solid foundation for clinical application and future research updates.

There is a notable deviation from the original study design outlined in the PROSPERO protocol. For instance, while the protocol anticipates the inclusion of RCTs, these were not considered in the final manuscript. Additionally, the primary outcome of prevalence, as stated in the protocol, is absent from the manuscript, calling into question the protocol's relevance to the study. The authors should revise their PROSPERO protocol to accurately reflect the changes made, detailing the rationale behind these modifications and the strategies employed to mitigate any resultant limitations.

Response: This study adhered to the protocol and we considered RCTs, however to address the PICO these needed at an arm with women with PCOS and an arm of controls without PCOS, both with no intervention. Conducting our extensive search and screening process, we found that no RCTs met these specific criteria. As a result, the studies ultimately included in our systematic review consisted solely of observational studies.

We have compared prevalence in the two groups by using odds ratios. We would like to clarify that the central question in the protocol was:

- ***What is the difference in the prevalence of pregnancy and birth complications between women with and without PCOS?***

Therefore, the primary objective was to examine the prevalence of pregnancy outcomes in those with PCOS relative to those without. Reporting the prevalence in each group separately would not directly address this study question. We aimed to focus on comparative analyses, which are essential for understanding the specific risks associated with PCOS in the context of these outcomes.

The authors are encouraged to reformat the presentation of results in the supplementary materials. The current approach of directly inserting Stata-generated outputs is suboptimal. Improved legibility, particularly concerning the font size in the supplementary figures, would greatly enhance the readability and interpretability of the data.

Response: Thank you for your constructive comments. In response to your feedback, we have carefully reformatted the supplementary figures to enhance their readability and interpretability.

The inclusion of studies published between 2003 and 2019 (n=22) appears to contribute to heterogeneity; how might the impact of COVID-19—such as infection during pregnancy or restricted access to healthcare during lockdowns—affect these findings? If this has not been considered, the authors should investigate the potential influences of the pandemic on the study outcomes.

Response: Thank you for your comment regarding the potential impact of the COVID-19 pandemic on our study findings. Given that the pandemic began at the end of 2019, we do not expect that the COVID-19 factors, such as infection during pregnancy or restricted access to healthcare during lockdowns, influenced the outcomes analysed in this review. The cumulative analysis of the impact of time over 2003-2019 showed limited influence on results and only impacted birthweight, suggesting the inclusion of studies over this time, did not contribute to heterogeneity.